# Late Fusion of Transformers for Sentiment Analysis of Code-Switched Data

**Gagan Sharma** and **R Chinmay** and **Raksha Sharma**

gagan_s@cs.iitr.ac.in and r_c@cs.iitr.ac.in and raksha.sharma@cs.iitr.ac.in

Indian Insitute of Technology Roorke

Roorkee, Uttarakhand, India

## Abstract

Code-switching is a common phenomenon in multilingual communities and is often used on social media. However, sentiment analysis of code-switched data is a challenging yet less explored area of research. This paper aims to develop a sentiment analysis system for code-switched data. In this paper, we present a novel approach combining two transformers using logits of their output and feeding them to a neural network for classification. We show the efficacy of our approach using two benchmark datasets, *viz.*, English-Hindi (En-Hi), and English-Spanish (En-Es) availed by Microsoft GLUECoS. Our approach results in an $F1$ score of 73.66% for En-Es and 61.24% for En-Hi, significantly higher than the best model reported for the GLUECoS benchmark dataset.

## 1 Introduction

Code-Switching refers to using two or more languages in a single conversation. This linguistic feature is prevalent in multilingual communities all over the world. Multilingual communities share vast amounts of unstructured code-switched data on social media in text and speech. Sentiment analysis of data shared on social media has proved valuable in market predictions and analysis (Darapaneni et al., 2022), mental health assessments (Calvo et al., 2017), and intelligent agents (Rambocas and Pacheco, 2018). There have been enormous studies conducted on multilingual NLP, but, despite such prevalence of code-switching in real life, very few studies have been conducted in the field.

Wang et al., (2018) proposed Generalized Language Evaluation Benchmark (GLUE) to evaluate embedding models on various language understanding tasks. Inspired by this, Microsoft Research published GLUECoS (Khanuja et al., 2020), an evaluation benchmark on various tasks focused on code-switched text. Sentiment Analysis is one of

| Hindi | Romanized |
|---|---|
| नया (*new*) | naya nya |
| खत्म (*end*) | khatam khtm khatm |
| बस (*sufficient*) | bas bs bus |

Figure 1: Hindi words with multiple possible Romanization

the seven tasks present in the GLUECoS evaluation benchmark, which is aimed at classifying sentiment in a code-switched sentence as *positive*, *negative*, or *neutral*. The task consists of English-Spanish and English-Hindi code-switched language pairs. The English-Spanish task is useful for studying the performance of models in a code-switched environment of closely related languages, and the English-Hindi task is for largely different languages.

The code-switched data analysis comes with its challenges. Traditional techniques trained for one language (monolingual NLP models) quickly break down when input is mixed in from another language. The performance of these models goes down as the amount of mixed-language present increases (Aguilar et al., 2019). On the other hand, there is a scarcity of high-quality code-switched datasets (Sitaram et al., 2020). Conversations on social media usually use slang, unknown references, mentions, improper grammar, spelling errors, *etc.* Models trained on correct grammar and spelling will not perform well on such datasets immediately (Dhingra et al., 2016).

In addition to this, there is also the case of multiple Romanization. A code switch statement is usually found in the Roman script because of standard English keyboards' vast popularity and availability. Multiple Romanization leads to words in foreign languages written in English alphabets with varying spellings, as shown in English-Hindi examples in Figure 1.

The cause of such different spellings is usually the difference in pronunciations which gets

reflected in the absence or addition of vowels in the Romanized foreign words.

**Our contribution** to the sentiment analysis of code-switched data research includes the performance improvement from the combination (late fusion) of different pre-trained models without using ensemble techniques. Since pre-trained models are trained on different datasets and have different architectures, they produce varying results on the same dataset. However, a unison model, which includes multiple pre-trained models, each fine-tuned on whole training data, improves performance compared to ensemble or other voting techniques. We show that our unisom model has improved the weighted-F1 score by 10% for the English-Spanish about 3.73% for the English-Hindi sentiment analysis task.

The rest of the paper is organized as follows. In Section 2, we briefly describe the details regarding the datasets used. Section 3 describes some recent related work. Section 4 describes our approach. Section 5 provides the experimental setup. Section 6 lists the results obtained, and Section 7 concludes the paper.

## 2 Dataset Description

GLUECoS is a reasonably significant evaluation benchmark for the six code-switched NLP tasks. A high performance sentiment analysis system for this dataset opens the scope for improvement in the performance of other code-switched NLP tasks. In this paper, we present results on the GLUECoS benchmark dataset. The dataset is generated by running a script that scrapes tweets of some pre-saved ids annotated beforehand (Khanuja et al., 2020). The English-Spanish dataset consists of 1523 tweets for training, 184 for validation, and 232 for testing. The English-Hindi dataset consists of 10080 tweets for training, 1260 for validation, and 1261 for testing. We have referred to the best results and method (mBERT) reported on the GLUECoS website[1] for the task. The objective of the paper is to improve upon the state-of-the-art method, that is, mBERT adapted for code-switched sentiment analysis.

Every data item is labeled to be expressing a *positive*, *negative*, and *neutral* sentiment. The detailed distribution of the annotations are listed in Table 1 and 2. Below is the definition of assigned labels.

---

[1]https://microsoft.github.io/GLUECoS/

- **Positive:** The statement infers joy, encouragement, likes, sarcasm, or excitement. For example, *I am amazed by the work done by the VFX team.*

- **Negative:** The statement infers sadness, discouragement, dislikes, sadism, or indifference. For example, *The food service was horrible at our hotel.*

- **Neutral:** The statement is mostly factual and does not demonstrate much of an opinion. For example, *The Taj Mahal is one of the World's Seven Wonders.*

| Sentiment | En-Es | En-Hi |
|---|---|---|
| Positive | 433 | 3202 |
| Negative | 379 | 2319 |
| Neutral | 711 | 4559 |
| **Total** | **1523** | **10080** |

Table 1: Training Set Distribution

| Sentiment | En-Es | En-Hi |
|---|---|---|
| Positive | 55 | 399 |
| Negative | 46 | 283 |
| Neutral | 83 | 578 |
| **Total** | **184** | **1260** |

Table 2: Validation Set Distribution

## 3 Literature Survey

Several research and industrial works have been done on sentiment analysis on textual data, especially in the previous decade. Recently, the research focused on code-switched data has been growing in popularity since a large share of posts available in social media or general conversations includes two or more languages (Winata et al., 2022). Before 2019, most sentiment analysis research revolved around techniques such as lexical sentiment analysis (like Bag-of-Words technique) (Kasthuriarachchy et al., 2014), semi-supervised or unsupervised algorithms (Maas et al., 2011), convolution models (Jianqiang et al., 2018), recurrent models (Arras et al., 2017), *etc*. These models failed to capture the context of the sentences and needed to be more accurate compared to transformers (Ran et al., 2018). Transformer-based models

replaced these classical models since the availability of labeled data was the main bottleneck for training neural networks (Devlin et al., 2019).

Semi-supervised algorithms such as Label Propagation (LP) algorithms (Wang et al., 2015) were used to predict sentiments in Code-switched data, using both statistical machine translation and sentiment information for words. In 2016, Wang et al., (2016) proposed a Bilingual Attention Network for code-switched emotion Prediction. They used Bi-LSTM to capture the context of words and both monolingual and bilingual attention vectors to predict the overall emotion of the input sentence. However, transformers (Vaswani et al., 2017) and pre-trained language models produced much better results than traditional techniques (Luo and Wang, 2019), with significantly less training data available. Fine-tuning the models with small amounts of available labeled data produced much better results than earlier methods. Recent research includes the generation of synthetic code-switched data (for better fine-tuning), intermediate task training (Prasad et al., 2021), continued pre-training (Li et al., 2022) (masked Language modeling using code-switched texts), late fusion (Mundra et al., 2021) (using multiple pre-trained transformer-based models and combining their outputs using some logic to improve overall performance), and Data Augmentation (Mundra et al., 2021) - both Random Augmentation (RA) and Balanced Augmentation (BA), and custom attention models (Li et al., 2021). Other techniques are used to improve sentiment analysis results in code-switched texts specific to language pairs or dataset specifications. In this paper, we propose a novel approach to enhance the late fusion technique using logits from models instead of the output.

## 4  Methodology

Due to the complementary behavior of different sentiment prediction models, we use the late fusion (Colneric and Demar, 2020) technique to combine results from different models as a unison model. The unison model, which includes multiple pre-trained models, each fine-tuned on whole training data, improves performance compared to ensemble or other voting techniques. We combine multiple models using their logit prediction for each sentiment. The description of such a model is given in Figure 2. The logit values for each sentiment class generated from two models are fed to a neural

network with three nodes in the output layer, one for each sentiment class. The neural network post-model training handles assigning weights to the results generated by each model. It thus gives more weight to the models as and when one performs better than the other and thus produce a new model which has the better aspects of each of the contributing model. The output neuron at the output layer with the highest value determines the sentiment class for the given input.

**Logit** is described as the output of a neuron without applying the activation function. Logit for a class is defined using the probability of the class being the predicted output. The mathematical definition of the logit is provided below.

$$Logit(p) = log(\frac{p}{(1-p)})$$

It is clear from the mathematical description that a larger logit value means a larger probability for the said sentiment class.

## 5  Experimental Setup

This section provides the experimental details to replicate the work.

### 5.1  Data Pre-processing

The dataset contains words that are written in different scripts. For the English-Hindi task, we need to transliterate words written in the Devanagari script to the Roman script to make it suitable for further processing. This is done using libindic's indic-trans (Bhat et al., 2015) library.

### 5.2  Hyper-parameters of the models

We have implemented **BERT** (Devlin et al., 2019), **mBERT**, **XLM-R** (Conneau et al., 2019) models in our work. For English-Spanish Task, the hyper-parameters are mentioned in Table 3. We used AdamW Optimizer (Loshchilov and Hutter, 2019) for all models. Other values are default values from the Hugging-Face library (refer BERT, mBERT, XLM-R).

| Model | gain | epsilon | batch | epoch |
|-------|------|---------|-------|-------|
| BERT  | 5e-5 | 1e-8    | 16    | 5     |
| mBERT | 4e-5 | 1e-8    | 16    | 5     |
| XLM-R | 3e-5 | 1e-8    | 16    | 5     |

Table 3: Hyper-parameters for En-Es models

Table 4 mentions the parameters for the English-Hindi task. For this task, we have used *Hi-BERT*

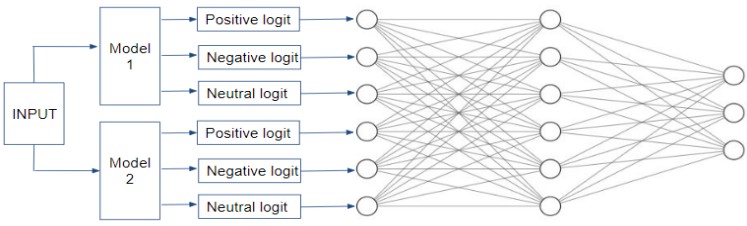

Figure 2: Late Fusion Neural Network

which is an ELECTRA model (Clark et al., 2020) trained on Hindi Wikipedia data and Hindi CommonCrawl dataset. For tokenization, we used *Bert-Tokenizer*. Other values are the default from the Hugging-Face library (refer Hi-BERT).

| Model | gain | epsilon | batch | epoch |
|--------|------|---------|-------|-------|
| BERT | 5e-5 | 1e-8 | 32 | 3 |
| mBERT | 4e-5 | 1e-8 | 32 | 3 |
| XLM-R | 3e-5 | 1e-8 | 16 | 5 |
| Hi-BERT | 5e-5 | 1e-8 | 32 | 3 |

Table 4: Hyper-parameters for En-Hi models

### 5.3 Evaluation Metric

We used the weighted $F1$ score as a metric of performance measure. The evaluation is done through a pull request at the GLUECoS GitHub repository. GitHub Actions run an evaluation script and generates the score.

### 5.4 Late Fusion

The two models generate logtis for each sentiment class. These values are fed to a neural network with six input neurons (Figure 2). Further, the model has one hidden layer with six hidden neurons and an output layer with three output neurons (one for each sentiment category). The neural network is created using the *MLPClassifier* of SKlearn. Other values are the default from the SKlearn implementation.[2]

### 6 Results

BERT has proven to be state-of-the-art for many NLP tasks, including sentiment analysis. For sentiment analysis of code-switched data, a multilingual version of BERT, that is, mBERT (Khanuja et al., 2020), is reported as state-of-the-art for the GLUE-CoS benchmark dataset. In this paper, we have tried to present a more accurate system by making an ensemble of multiple BERT-based models

---

[2]refer MLPClassifier.

| Model | F1 |
|--------|-----|
| mBERT (Khanuja et al., 2020) | 63.02% |
| BERT | 65.05% |
| XLM-R | 69.42% |
| mBERT - BERT (Late Fusion) | 69.40% |
| XLM-R - BERT (Late Fusion) | 70.9% |
| **XLM-R - mBERT** (Late Fusion) | **73.66%** |

Table 5: F1 Scores Obtained for English-Spanish Sentiment Analysis.

suitable for multiple languages text (*viz.*, mBERT, XLM-R, Hi-BERT).[3] Tables 5 and 6 present the results of our approach and various other methods for English-Spanish and English-Hindi code-switched dataset for sentiment analysis. Khanuja et al., (2020) presented *mBERT* as these datasets' best-performing model. We further produced results with *BERT*, *XLM-R* for both the datasets and *Hi-BERT* for the English-Hindi dataset. The results obtained from the late fusion of two models are better than those obtained from individual models. In our late fusion approach, the neural network post-model training handles assigning weights to the results generated by each model. It thus gives more weight to the models as and when one performs better than the other and thus produce a new model which has the better aspects of each of the contributing model. The late fusion technique reduces the predictions' dispersion, making the end model more robust. We obtained the best F1 score of 73.66% for English-Spanish with the late fusion of XLM-R and mBERT. While for English-Hindi, mBERT and BERT produced the best F1 score of 61.24%.

---

[3]The majority voting-based ensemble method results were worse than the independent use of BERT/mBERT/XLM-R models. Hence, we focused on showing improvement over the individual model.

| Model | F1 |
|---|---|
| XLM-R | 28.5% |
| Hi-BERT | 55.90% |
| mBERT (Khanuja et al., 2020) | 57.51% |
| BERT | 59.05% |
| BERT - Hi-BERT (Late Fusion) | 59.73% |
| mBERT - Hi-BERT (Late Fusion) | 59.97% |
| **mBERT - BERT** (Late Fusion) | **61.24%** |

Table 6: F1 Scores Obtained for English-Hindi Sentiment Analysis.

## 7 Conclusion

This paper presents a novel approach that uses late fusion for sentiment analysis of code-switched data. The results show that *mBERT-BERT* and *XLMR-mBERT* fusion models perform the best for English-Hindi and English-Spanish Sentiment Analysis tasks, respectively. The tools and models used in our work are largely non-language specific and thus could be helpful in Sentiment Analysis of code-switched data of different languages.

## Limitations

This paper presents results for code-switched data in English-Hindi and English-Spanish. The languages involved in the work are widely spoken languages across the world over social media. However, we have not observed the performance of our fusion-based approach for code-switched data where one of the languages is low resource language such as English and Dravidian (Malayalam, Tamil, Kannada). It requires further investigation and experimentation.

## Ethics Statement

The work reported in the paper complies with the ACL Ethics Policy. The output of our work does not have any kind of negative impact on society; it is for the good of society.

## Acknowledgements

We are grateful to Dr. Rudra Murthy from IBM Research Bangalore for his guidance throughout the execution of the project.

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
