# OpenReview forum: "Late Fusion of Transformers for Sentiment Analysis of Code-Switched Data"
_EMNLP/2023/Conference — EMNLP 2023 Findings_

### Official Review · Reviewer_L6it · 2023-08-02

**Soundness:** 4

**Excitement:**

4: Strong: This paper deepens the understanding of some phenomenon or lowers the barriers to an existing research direction.

**Paper Topic And Main Contributions:**

The authors in this paper presented an approach for Sentiment Analysis of Code-Switched Data. They combined two transformers using logits of their output and feeding them to a neural network for classification. They evaluated the efficacy of their approach using two benchmark datasets of English-Hindi and English-Spanish.

**Reasons To Accept:**

The concept of the paper is based on an innovative method. They introduce a new fusion technique instead of an ensemble majority voting model.

**Reasons To Reject:**

An addition that could be done is a conventional majority voting technique and comparing it with their late fusion technique.

**Reproducibility:**

4: Could mostly reproduce the results, but there may be some variation because of sample variance or minor variations in their interpretation of the protocol or method.

**Reviewer Confidence:**

5: Positive that my evaluation is correct. I read the paper very carefully and I am very familiar with related work.

---

> ### Author Rebuttal · Authors · 2023-08-28
>
> Thanks a lot for reading the paper in depth. Due to the limitation of 4 pages, we had to confine the information in all sections, including the results section. We want to provide more clarification with the help of the following points through rebuttal. We request the reviewer to increase the score of the paper considering the clarification provided in the rebuttal.
>
> - The majority voting-based ensemble method results were worse than the independent use of BERT/mBERT/XLM-R models. Hence, we focused on showing improvement over the individual model.

---

### Official Review · Reviewer_FBU8 · 2023-08-03

**Soundness:** 3

**Excitement:**

3: Ambivalent: It has merits (e.g., it reports state-of-the-art results, the idea is nice), but there are key weaknesses (e.g., it describes incremental work), and it can significantly benefit from another round of revision. However, I won't object to accepting it if my co-reviewers champion it.

**Paper Topic And Main Contributions:**

The authors present a sentiment analysis classifier for code-switched (mixed languages) datasets. They leverage two public datasets, one with English & Spanish and another with English-Hindi. Their approach involves combining two state-of-the-art transformers models by training a third model responsible for merging their output classification, a Late Fusion technique. Their solution presents a better performance in terms of F1 score compared to other single-model approaches.

**Reasons To Accept:**

- The authors presents a clear and direct approach to solve the proposed problem of sentiment analysis in code-switched text.
- Their approach sounds reproducible and uses a well known public available data. Their methodology seems to be reproducible with proper hyper-parameters discussion.
- Not just their solution outperforms their baseline, but the proposed approach is easy to understand and well justified for the problem.

**Reasons To Reject:**

- The paper sounds like an incremental traditional sentiment analysis paper without anything new. One may expect a solution that could leverage the context of the problem to solve it instead of just combining two models' outputs to outperform a selected baseline.
- It needs to be clarified why not using ensemble techniques is an advantage. Explain it.
- In the literature review, you state that data availability was the bottleneck for transformers, but in the following paragraph, you stated that transformers produced better results than traditional techniques with significantly less training data. These sentences are contradictory.

**Reproducibility:**

4: Could mostly reproduce the results, but there may be some variation because of sample variance or minor variations in their interpretation of the protocol or method.

**Reviewer Confidence:**

4: Quite sure. I tried to check the important points carefully. It's unlikely, though conceivable, that I missed something that should affect my ratings.

**Typos Grammar Style And Presentation Improvements:**

-The text would greatly be enriched if you share in the text examples of code-switch cases.
-"availed"?
- unison vs unisom

---

> ### Author Rebuttal · Authors · 2023-08-28
>
> Due to the limitation of 4 pages, we had to confine the information. We want to provide more clarification with the help of the following points through rebuttal. We will incorporate these details in the camera-ready version. We request the reviewer to increase the score of the paper considering the clarification provided in the rebuttal.
>
> 1. mBERT is a BERT model suitable to process datasets having multiple language words. GLUECoS is a significant evaluation benchmark for the six code-switched NLP tasks. We have referred to the best results and method (mBERT) reported on the GLUECoS website for the task: https://microsoft.github.io/GLUECoS/. The paper's objective is to improve upon the SOTA method, that is, mBERT adapted for code-switched sentiment analysis. A high-performance sentiment analysis system for this dataset opens the scope for improvement in the performance of other code-switched NLP tasks.
>
> 2. Line 080 - 'Since pre-trained models are trained on different datasets and have different architectures, they produce varying results on the same dataset. However, a unison model, which includes multiple pre-trained models, each fine-tuned on whole training data, improves performance compared to ensemble or other voting techniques.'
>
> 3. The majority voting-based ensemble method results were worse than the independent use of BERT/mBERT/XLM-R models. Hence, we focused on showing improvement over the individual model.
>
> 4. Line 141: 'Transformer-based models replaced these classical models since the availability of labeled data was the main bottleneck for training neural networks (Devlin et al., 2019).' The statement suggests there needed to be more labeled data in the past to utilize neural networks with full capacity. However, at the current time, we can deploy transformers with total capacity due to the availability of a significant amount of labeled datasets.

---

### Official Review · Reviewer_ZSXs · 2023-08-04

**Soundness:** 2

**Excitement:**

2: Mediocre: This paper makes marginal contributions (vs non-contemporaneous work), so I would rather not see it in the conference.

**Missing References:**

- SemEval-2020 Task 9: Overview of Sentiment Analysis of Code-Mixed Tweets (Patwa et al., SemEval 2020). Also, the participating teams are relevant.

**Paper Topic And Main Contributions:**

The paper targets the problem of sentiment analysis for code-switch data. The paper provides an overview of the sentiment analysis task and focuses on the challenges that arise when dealing with code-switch data, such as multiple Romanizations of the same word. The authors aim to improve the performance on sentiment analysis of code switch data. This is done through a model based on late fusion, where multiple models' output is combined. The hypothesis is that since pre-trained language models are trained on varying data, they would provide different results on the same dataset. In this work, the authors fine-tune two models for the task and then utilise their feed their logits to a feed-forward network, which produces the final prediction. The authors experimented with the GLUECos benchmark datasets, specifically the English-Spanich and English-Hindi datasets.

**Questions For The Authors:**

- The reported results do not match the ones in the original CLEUCos paper. Can you elaborate on why this is the case?
- It seems that you first fine-tune the models on the whole dataset and then utilise the fusion. This was not mentioned clearly in the methodology; can you confirm that?
- Can you justify why you only experimented with three models?
- Can you justify using the suggested dataset? Why did you not include other datasets, such as SentiMix?
- Is there any more recent works that report on the SAIL dataset (the original CLEUCos dataset)?


**Reasons To Accept:**

- The paper shows that there is an improvement on the used dataset.
- The approach seems reproducible.

**Reasons To Reject:**

- The paper lacks enough empirical evidence of the effectiveness of the proposed model. Since the main contribution of the paper is the model, there should be more experiments to support that. To achieve this, it is suggested to experiment with more models and test on other datasets.
- The paper reports on a single dataset and compares the performance against one model from 2020.
- The paper lacks some clarity in explaining the methodology. It is not clear how the models are fine-tuned before using the fusion.


**Reproducibility:**

4: Could mostly reproduce the results, but there may be some variation because of sample variance or minor variations in their interpretation of the protocol or method.

**Reviewer Confidence:**

4: Quite sure. I tried to check the important points carefully. It's unlikely, though conceivable, that I missed something that should affect my ratings.

**Typos Grammar Style And Presentation Improvements:**

- The methodology section needs improvement. It is not clear that you fine-tune the models separately, and then utilise them for the final model.
- lines 83, 87 (unison vs unisom).

---

> ### Author Rebuttal · Authors · 2023-08-28
>
> Thanks a lot for reading the paper thoroughly and providing valuable comments. Due to the limitation of 4 pages, we had to confine the information. We want to provide more clarification with the help of the following points through rebuttal. We will incorporate these details in the camera-ready version. We request the reviewer to increase the score of the paper considering the clarification provided in the rebuttal.
>
> 1. The reported results do not match the ones in the original GCLEUCos paper. Can you elaborate on why this is the case?
>
> - We have referred to the best results and method (mBERT) reported on the GLUECoS website for the task: https://microsoft.github.io/GLUECoS/. The objective of the paper is to improve upon the SOTA method, that is, mBERT adapted for code-switched sentiment analysis.
>
> 2. It seems that you first fine-tune the models on the whole dataset and then utilise the fusion. This was not mentioned clearly in the methodology; can you confirm that?
>
> - Yes, please see the ‘Our contribution’ section at line number – 083.
>
> 3. Can you justify why you only experimented with three models?
>
> - BERT has proven to be state-of-the-art for many NLP tasks, including sentiment analysis. For sentiment analysis of code-switched data, a multilingual version of BERT, that is, mBERT (Khanuja et al., 2020), is reported as state-of-the-art for the GLUECoS benchmark dataset. In this paper, we have tried to present a more accurate system by making an ensemble of multiple BERT-based models suitable for multiple languages text  (viz., mBERT, XLM-R, Hi-BERT). We have shown a comparison among six systems (Table 5) for English-Spanish setup and seven systems (Table 6) for English-Hindi setup.  We believe this suffices the list of experiments to support the paper's objective.
>
> 4. Can you justify using the suggested dataset? Why did you not include other datasets, such as SentiMix?
>
> - GLUECoS is a reasonably significant evaluation benchmark for the six code-switched NLP tasks. A high performance sentiment anlysis system for this dataset opens the scope for improvement in the performance of other code-switched NLP tasks.
>
> 5. Is there any more recent works that report on the SAIL dataset (the original CLEUCos dataset)?
>
> - No, to the best of our knowledge.

---

### Meta-Review · Area_Chair_u6iP · 2023-09-26

**Recommendation:** 3

**Metareview:**

This paper addresses sentiment analysis for code-switched data, emphasizing the challenges posed by mixed languages. The authors propose a late fusion model that combines the outputs of two fine-tuned transformers and demonstrates improved performance on English-Spanish and English-Hindi datasets.

Reasons to accept:
The paper demonstrates a clear improvement on the dataset used for sentiment analysis.
The approach is presented in a clear and direct manner, making it easy to understand and reproduce.
The methodology is reproducible, utilizing well-known and publicly available data with proper hyperparameter discussion.
The proposed approach not only outperforms the baseline but is also well-justified for the problem at hand.

Reasons to reject:
- Insufficient empirical evidence to support the effectiveness of the proposed model, with a lack of experiments involving additional models and diverse datasets.
- Limited focus on a single dataset and comparison against a single 2020 model, raising concerns about generalizability.
-Lack of clarity in explaining the methodology, particularly in terms of how the models are fine-tuned before fusion.
-The paper is perceived as an incremental traditional sentiment analysis work without introducing novel solutions that leverage context for problem-solving.
-The advantage of not using ensemble techniques is not adequately explained.
-Contradictory statements in the literature review regarding data availability and transformer performance with less training data.

---

### Decision · Program_Chairs · 2023-10-07

**Decision:**

Accept-Findings

**Comment:**

This paper addresses sentiment analysis for code-switched data, emphasizing the challenges posed by mixed languages. The authors propose a late fusion model that combines the outputs of two fine-tuned transformers and demonstrates improved performance on English-Spanish and English-Hindi datasets.

Reasons to accept:
The paper demonstrates a clear improvement on the dataset used for sentiment analysis.
The approach is presented in a clear and direct manner, making it easy to understand and reproduce.
The methodology is reproducible, utilizing well-known and publicly available data with proper hyperparameter discussion.
The proposed approach not only outperforms the baseline but is also well-justified for the problem at hand.

Reasons to reject:
- Insufficient empirical evidence to support the effectiveness of the proposed model, with a lack of experiments involving additional models and diverse datasets.
- Limited focus on a single dataset and comparison against a single 2020 model, raising concerns about generalizability.
-Lack of clarity in explaining the methodology, particularly in terms of how the models are fine-tuned before fusion.
-The paper is perceived as an incremental traditional sentiment analysis work without introducing novel solutions that leverage context for problem-solving.
-The advantage of not using ensemble techniques is not adequately explained.
-Contradictory statements in the literature review regarding data availability and transformer performance with less training data.